# Diverse Randomized Agents Vote to Win

**Albert Xin Jiang**
Trinity University
xjiang@trinity.edu

**Leandro Soriano Marcolino**
USC
sorianom@usc.edu

**Ariel D. Procaccia**
CMU
arielpro@cs.cmu.edu

**Tuomas Sandholm**
CMU
sandholm@cs.cmu.edu

**Nisarg Shah**
CMU
nkshah@cs.cmu.edu

**Milind Tambe**
USC
tambe@usc.edu

## Abstract

We investigate the power of voting among diverse, randomized software agents. With teams of computer Go agents in mind, we develop a novel theoretical model of two-stage noisy voting that builds on recent work in machine learning. This model allows us to reason about a collection of agents with different biases (determined by the first-stage noise models), which, furthermore, apply randomized algorithms to evaluate alternatives and produce votes (captured by the second-stage noise models). We analytically demonstrate that a uniform team, consisting of multiple instances of any single agent, must make a significant number of mistakes, whereas a diverse team converges to perfection as the number of agents grows. Our experiments, which pit teams of computer Go agents against strong agents, provide evidence for the effectiveness of voting when agents are diverse.

## 1 Introduction

Recent years have seen a surge of work at the intersection of social choice and machine learning. In particular, significant attention has been given to the learnability and applications of *noisy preference models* [16, 2, 1, 3, 24]. These models enhance our understanding of voters' behavior in elections, and provide a theoretical basis for reasoning about crowdsourcing systems that employ voting to aggregate opinions [24, 8]. In contrast, this paper presents an application of noisy preference models to the design of systems of *software agents*, emphasizing the importance of voting and diversity.

Our starting point is two very recent papers by Marcolino et al. [19, 20], which provide a new perspective on voting among multiple software agents. Their empirical results focus on Computer Go programs (see, e.g., [10]), which often use Monte Carlo tree search algorithms [7]. Taking the team formation point of view, Marcolino et al. establish that a team consisting of multiple (four to six) different computer Go programs that use plurality voting — each agent giving one point to a favorite alternative — to decide on each move outperforms a team consisting of multiple copies of the strongest program (which is better than a single copy because the copies are initialized with different random seeds). The insight is that even strong agents are likely to make poor choices in some states, which is why diversity beats strength. And while the benefits of diversity in problem solving are well studied [12, 13, 6, 14], the setting of Marcolino et al. combines several ingredients. First, performance is measured across multiple states; as they point out, this is also relevant when making economic decisions (such as stock purchases) across multiple scenarios, or selecting item recommendations for multiple users. Second, agents' votes are based on randomized algorithms; this is also a widely applicable assumption, and in fact even Monte Carlo tree search specifically is used for problems ranging from traveling salesman to classical (deterministic) planning, not to mention that randomization is often used in many other AI applications.

Focusing on the computer Go application, we find it exciting because it provides an ideal example of voting among teams of software agents: It is difficult to compare quality scores assigned by heterogeneous agents to different moves, so optimization approaches that rely on *cardinal utilities* fall short while voting provides a natural aggregation method. More generally the setting's new ingredients call for a novel model of social choice, which should be rich enough to explain the empirical finding that diversity beats strength.

However, the model suggested by Marcolino et al. [19] is rather rudimentary: they prove that a diverse team would outperform copies of the strongest agent *only if* one of the weaker agents outperforms the strongest agent in at least one state; their model cannot quantify the advantage of diversity. Marcolino et al. [20] present a similar model, but study the effect of increasing the size of the action space (i.e., the board size in the Go domain). More importantly, Marcolino et al. [19, 20] — and other related work [6] — assume that each agent votes for a single alternative. In contrast, it is potentially possible to design agents that generate a ranking of multiple alternatives, calling for a principled way to harness this additional information.

## 1.1 Our Approach and Results

We introduce the following novel, abstract model of voting, and instantiate it using Computer Go. In each state, which corresponds to a board position in Go, there is a ground truth, which captures the true quality of different alternatives — feasible moves in Go. Heuristic agents have a noisy perception of the quality of alternatives. We model this using a *noise model* for each agent, which randomly maps the ground truth to a ranking of the alternatives, representing the agent's *biased* view of their qualities. But if a single agent is presented with the same state twice, the agent may choose two different alternatives. This is because agents are assumed to be randomized. For example, as mentioned above, most computer Go programs, such as Fuego [10], rely on Monte Carlo Tree Search to randomly decide between different moves. We model this additional source of noise via a second noise model, which takes the biased ranking as input, and outputs the agent's *vote* (another ranking of the alternatives). A *voting rule* is employed to select a single alternative (possibly randomly) by aggregating the agents' votes. Our main theoretical result is the following theorem, which is, in a sense, an extension of the classic Condorcet Jury Theorem [9].

**Theorem 2 (simplified and informal).** *(i) Under extremely mild assumptions on the noise models and voting rule, a* uniform team *composed of copies of any single agent (even the "strongest" one with the most accurate noise models), for any number of agents and copies, is likely to vote for suboptimal alternatives in a significant fraction of states; (ii) Under mild assumptions on the noise models and voting rule, a* diverse team *composed of a large number of different agents is likely to vote for optimal alternatives in almost every state.*

We show that the assumptions in both parts of the theorem are indeed mild by proving that three well-known noise models — the Mallows-$\phi$ model [18], The Thurstone-Mosteller model [26, 21], and the Plackett-Luce model [17, 23] — satisfy the assumptions in both parts of the theorem. Moreover, the assumptions on the voting rule are satisfied by almost all prominent voting rules.

We also present experimental results in the Computer Go domain. As stated before, our key methodological contributions are a procedure for automatically generating diverse teams by using different parameterizations of a Go program, and a novel procedure for extracting rankings of moves from algorithms that are designed to output only a single good move. We show that the diverse team significantly outperforms the uniform team under the plurality rule. We also show that it is possible to achieve better performance by extracting rankings from agents using our novel methodology, and aggregating them via ranked voting rules.

## 2 Background

We use $[k]$ as shorthand for $\{1, \ldots, k\}$. A vote is a total order (ranking) over the alternatives, usually denoted by $\sigma$. The set of rankings over a set of alternatives $A$ is denoted by $\mathcal{L}(A)$. For a ranking $\sigma$, we use $\sigma(i)$ to denote the alternative in position $i$ in $\sigma$, so, e.g., $\sigma(1)$ is the most preferred alternative in $\sigma$. We also use $\sigma([k])$ to denote $\{\sigma(1), \ldots, \sigma(k)\}$. A collection of votes is called a *profile*, denoted by $\pi$. A *deterministic voting rule* outputs a winning alternative on each profile. For a *randomized voting rule* $f$ (or simply a voting rule), the output $f(\pi)$ is a distribution over the alternatives. A

voting rule is *neutral* if relabeling the alternatives relabels the output accordingly; in other words, the output of the voting rule is independent of the labels of the alternatives. All prominent voting rules, when coupled with uniformly random tie breaking, are neutral.

**Families of voting rules.** Next, we define two families of voting rules. These families are quite wide, disjoint, and together they cover almost all prominent voting rules.

- *Condorcet consistency.* An alternative is called the Condorcet winner in a profile if it is preferred to every other alternative in a majority of the votes. Note that there can be at most one Condorcet winner. A voting rule is called *Condorcet consistent* if it outputs the Condorcet winner (with probability 1) whenever it exists. Many famous voting rules such as Kemeny's rule, Copeland's rule, Dodgson's rule, the ranked pairs method, the maximin rule, and Schulze's method are Condorcet consistent.

- *PD-c Rules [8].* This family is a generalization of positional scoring rules that include prominent voting rules such as plurality and Borda count. While the definition of Caragiannis et al. [8] outputs rankings, we naturally modify it to output winning alternatives.

  Let $T_\pi(k, a)$ denote the number of times alternative $a$ appears among first $k$ positions in profile $\pi$. Alternative $a$ is said to *position-dominate* alternative $b$ in $\pi$ if $T_\pi(k, a) > T_\pi(k, b)$ for all $k \in [m - 1]$, where $m$ is the number of alternatives in $\pi$. An alternative is called the *position-dominating winner* if it position-dominates every other alternative in a profile. It is easy to check that there can be at most one position-dominating winner. A voting rule is called *position-dominance consistent* (PD-c) if it outputs the position-dominating winner (with probability 1) whenever it exists. Caragiannis et al. [8] show that all positional scoring rules (including plurality and Borda count) and Bucklin's rule are PD-c (as rules that output rankings). We show that this holds even when the rules output winning alternatives. This is presented as Proposition 1 in the online appendix (specifically, Appendix A).

Caragiannis et al. [8] showed that PD-c rules are disjoint from Condorcet consistent rules (actually, for rules that output rankings, they use a natural generalization of Condorcet consistent rules that they call PM-c rules). Their proof also establishes the disjointness of the two families for rules that output winning alternatives.

## 2.1 Noise Models

One view of computational social choice models the votes as noisy estimates of an unknown true order of the alternatives. These votes come from a distribution that is parametrized by some underlying ground truth. The ground truth can itself be the true order of alternatives, in which case we say that the noise model is of the *rank-to-rank* type. The ground truth can also be an objective true quality level for each alternative, which is more fine-grained than a true ranking of alternatives. In this case, we say that the noise model is of the *quality-to-rank* type. See [15] for examples of quality-to-rank models and how they are learned. Note that the output votes are rankings over alternatives in both cases. We denote the ground truth by $\boldsymbol{\theta}$. It defines a *true ranking* of the alternatives (even when the ground truth is a quality level for each alternative), which we denote by $\sigma^*$.

Formally, a noise model $P$ is a set of distributions over rankings — the distribution corresponding to the ground truth $\boldsymbol{\theta}$ is denoted by $P(\boldsymbol{\theta})$. The probability of sampling a ranking $\sigma$ from $P(\boldsymbol{\theta})$ is denoted by $\Pr_P[\sigma; \boldsymbol{\theta}]$.

Similarly to voting rules, a noise model is called *neutral* if relabeling the alternatives permutes the probabilities of various rankings accordingly. Formally, a noise model $P$ is called neutral if $\Pr_P[\sigma; \boldsymbol{\theta}] = \Pr_P[\tau\sigma; \tau\boldsymbol{\theta}]$, for every permutation $\tau$ of the alternatives, every ranking $\sigma$, and every ground truth $\boldsymbol{\theta}$. Here, $\tau\sigma$ and $\tau\boldsymbol{\theta}$ denote the result of applying $\tau$ on $\sigma$ and $\boldsymbol{\theta}$, respectively.

**Classic noise models.** Below, we define three classical noise models:

- *The Mallows-$\phi$ model* [18]. This is a rank-to-rank noise model, where the probability of a ranking decreases exponentially in its distance from the true ranking. Formally, the Mallows-$\phi$ model for $m$ alternatives is defined as follows. For all rankings $\sigma$ and $\sigma^*$,

$$\Pr[\sigma; \sigma^*] = \frac{\phi^{d_{KT}(\sigma, \sigma^*)}}{Z_\phi^m}, \tag{1}$$

where $d_{KT}$ is the Kendall-Tau distance that measures total pairwise disagreement between two rankings, and the normalization constant $Z_\phi^m = \prod_{k=1}^m \sum_{j=0}^{k-1} \phi^j$ is independent of $\sigma^*$.

- *The Thurstone-Mosteller (TM)* [26, 21] and the *Plackett-Luce (PL)* [17, 23] models. Both models are of the quality-to-rank type, and are special cases of a more general *random utility model* (see [2] for its use in social choice). In a random utility model, each alternative $a$ has an associated true quality parameter $\theta_a$ and a distribution $\mu_a$ parametrized by $\theta_a$. In each sample from the model, a noisy quality estimate $X_a \sim \mu_a(\theta_a)$ is obtained, and the ranking where the alternatives are sorted by their noisy qualities is returned.

  For the Thurstone-Mosteller model, $\mu_a(\theta_a)$ is taken to be the normal distribution $\mathcal{N}(\theta_a, \nu^2)$ with mean $\theta_a$, and variance $\nu^2$. Its PDF is

  $$f(x) = \frac{1}{\sqrt{2\pi\nu^2}} e^{-\frac{(x-\theta_a)^2}{2\nu^2}}.$$

  For the Plackett-Luce model, $\mu_a(\theta_a)$ is taken to be the Gumbel distribution $\mathcal{G}(\theta_a)$. Its PDF follows $f(x) = e^{-(x-\theta_a) - e^{-(x-\theta_a)}}$. The CDF of the Gumbel distribution $\mathcal{G}(\theta_a)$ is given by $F(x) = e^{-e^{-(x-\theta_a)}}$. Note that we do not include a variance parameter because this subset of Gumbel distributions is sufficient for our purposes.

  The Plackett-Luce model has an alternative, more intuitive, formulation. Taking $\lambda_a = e^{\theta_a}$, the probability of obtaining a ranking is the probability of sequentially choosing its alternatives from the pool of remaining alternatives. Each time, an alternative is chosen among a pool proportional to its $\lambda$ value. Hence, $\Pr[\sigma; \{\lambda_a\}] = \prod_{i=1}^m \frac{\lambda_{\sigma(i)}}{\sum_{j=i}^m \lambda_{\sigma(j)}}$, where $m$ is the number of alternatives.

## 3 Theoretical Results

In this section, we present our theoretical results. But, first, we develop a novel model that will provide the backdrop for these results. Let $N = \{1, \ldots, n\}$ be a set of agents. Let $S$ be the set of states of the world, and let $|S| = t$. These states represent different scenarios in which the agents need to make decisions; in Go, these are board positions. Let $\mu$ denote a probability distribution over states in $S$, which represents how likely it is to encounter each state. Each state $s \in S$ has a set of alternatives $A_s$, which is the set of possible actions the agents can choose in state $s$. Let $|A_s| = m_s$ for each $s \in S$. We assume that the set of alternatives is fixed in each state. We will later see how our model and results can be adjusted for varying sets of alternatives. The ground truth in state $s \in S$ is denoted by $\boldsymbol{\theta}_s$, and the true ranking in state $s$ is denoted by $\sigma_s^*$.

**Votes of agents.** The agents are presented with states sampled from $\mu$. Their goal is to choose the true best alternative, $\sigma_s^*(1)$, in each state $s \in S$ (although we discuss why our results also hold when the goal is to maximize expected quality). The inability of the agents to do so arises from two different sources: the suboptimal heuristics encoded within the agents, and their inability to fully optimize according to their own heuristics — these are respectively modeled by two noise models $P_i^1$ and $P_i^2$ associated with each agent $i$.

The agents inevitably employ heuristics (in domains like Go) and therefore can only obtain a noisy evaluation of the quality of different alternatives, which is modeled by the noise model $P_i^1$ of agent $i$. The biased view of agent $i$ for the true order of the alternatives in $A_s$, denoted $\sigma_{is}$, is modeled as a sample from the distribution $P_i^1(\sigma_s^*)$. Moreover, we assume that the agents' decision making is randomized. For example, top computer Go programs use Monte Carlo tree search algorithms [7]. We therefore assume that each agent $i$ has another associated noise model $P_i^2$ such that the final ranking that the agent returns is a sample from $P_i^2(\sigma_{is})$. To summarize, agent $i$'s vote is obtained by first sampling its biased truth from $P_i^1$, and then sampling its vote from $P_i^2$. It is clear that the composition $P_i^2 \circ P_i^1$ plays a crucial role in this process.

**Agent teams.** Since the agents make errors in estimating the best alternative, it is natural to form a team of agents and aggregate their votes. We consider two team formation methods: a *uniform* team comprising of multiple copies of a single agent that share the same biased truths but have different final votes due to randomness; and a *diverse* team comprising of a single copy of each agent with different biased truths and different votes. We show that the diverse team outperforms the uniform team irrespective of the choice of the agent that is copied in the uniform team.

### 3.1 Restrictions on Noise Models

No team can perform well if the noise models $P_i^1$ and $P_i^2$ lose all useful information. Hence, we impose intuitive restrictions on the noise models; our restrictions are mild, as we demonstrate (Theorem 1) that the three classical noise models presented in Section 2.1 satisfy all our assumptions.

**PM-$\alpha$ Noise Model** For $\alpha > 0$, a neutral noise model $P$ is called *pairwise majority preserving with strength $\alpha$* (or PM-$\alpha$) if for every ground truth $\boldsymbol{\theta}$ (and the corresponding true ranking $\sigma^*$) and every $i < j$, we have

$$\Pr_{\sigma \sim P(\boldsymbol{\theta})}[\sigma^*(i) \succ_\sigma \sigma^*(j)] \geq \Pr_{\sigma \sim P(\boldsymbol{\theta})}[\sigma^*(j) \succ_\sigma \sigma^*(i)] + \alpha, \tag{2}$$

where $\succ_\sigma$ is the preference relation of a ranking $\sigma$ sampled from $P(\boldsymbol{\theta})$. Note that this definition applies to both quality-to-rank and rank-to-rank noise models. In other words, in PM-$\alpha$ noise models every pairwise comparison in the true ranking is preserved in a sample with probability at least $\alpha$ more than the probability of it not being preserved.

**PD-$\alpha$ Noise Model** For $\alpha > 0$, a neutral noise model is called *position-dominance preserving with strength $\alpha$* (or PD-$\alpha$) if for every ground truth $\boldsymbol{\theta}$ (and the corresponding true ranking $\sigma^*$), every $i < j$, and every $k \in [m-1]$ (where $m$ is the number of alternatives),

$$\Pr_{\sigma \sim P(\boldsymbol{\theta})}[\sigma^*(i) \in \sigma([k])] \geq \Pr_{\sigma \sim P(\boldsymbol{\theta})}[\sigma^*(j) \in \sigma([k])] + \alpha. \tag{3}$$

That is, for every $k \in [m-1]$, an alternative higher in the true ranking has probability higher by at least $\alpha$ of appearing among the first $k$ positions in a vote than an alternative at a lower position in the true ranking.

**Compositions of noise models with restrictions.** As mentioned above, compositions of noise models play an important role in our work. The next lemma shows that our restrictions on noise models are preserved, in a sense, under composition; its proof appears in Appendix B.

**Lemma 1.** *For $\alpha_1, \alpha_2 > 0$, the composition of a PD-$\alpha_1$ noise model with a PD-$\alpha_2$ noise model is a PD-$(\alpha_1 \cdot \alpha_2)$ noise model.*

Unfortunately, a similar result does not hold for PM-$\alpha$ noise models; the composition of a PM-$\alpha_1$ noise model and a PM-$\alpha_2$ noise model may yield a noise model that is not PM-$\alpha$ for any $\alpha > 0$. In Appendix C, we give such an example. While this is slightly disappointing, we show that a stronger assumption on the first noise model in the composition suffices.

**PPM-$\alpha$ Noise Model** For $\alpha > 0$, a neutral noise model $P$ is called *positional pairwise majority preserving* (or PPM-$\alpha$) if for every ground truth $\boldsymbol{\theta}$ (and the corresponding true ranking $\sigma^*$) and every $i < j$, the quantity

$$\Pr_{\sigma \sim P(\boldsymbol{\theta})}[\sigma(i') = \sigma^*(i) \wedge \sigma(j') = \sigma^*(j)] - \Pr_{\sigma \sim P(\boldsymbol{\theta})}[\sigma(j') = \sigma^*(i) \wedge \sigma(i') = \sigma^*(j)] \tag{4}$$

is non-negative for every $i' < j'$, and at least $\alpha$ for some $i' < j'$. That is, for $i' < j'$, the probability that $\sigma^*(i)$ and $\sigma^*(j)$ go to positions $i'$ and $j'$ respectively in a vote should be at least as high as the probability of them going to positions $j'$ and $i'$ respectively (and at least $\alpha$ greater for some $i'$ and $j'$). Summing Equation (4) over all $i' < j'$ shows that every PPM-$\alpha$ noise model is also PM-$\alpha$.

**Lemma 2.** *For $\alpha_1, \alpha_2 > 0$, if noise models $P^1$ and $P^2$ are PPM-$\alpha_1$ and PM-$\alpha_2$, respectively, then their composition $P^2 \circ P^1$ is PM-$(\alpha_1 \cdot \alpha_2)$.*

The lemma's proof is relegated to Appendix D.

### 3.2 Team Formation and the Main Theoretical Result

Let us explain the process of generating votes for the uniform team and for the diverse team. Consider a state $s \in S$. For the uniform team consisting of $k$ copies of agent $i$, the biased truth $\sigma_{is}$ is drawn from $P_i^1(\boldsymbol{\theta}_s)$, and is common to all the copies. Each copy $j$ then individually draws a vote $\pi_{is}^j$ from $P_i^2(\sigma_{is})$; we denote the collection of these votes by $\boldsymbol{\pi}_{is}^k = (\pi_{is}^1, \ldots, \pi_{is}^k)$. Under a voting rule $f$, let $X_{is}^k = \mathbb{I}[f(\boldsymbol{\pi}_{is}^k) = \sigma_s^*(1)]$ be the indicator random variable denoting whether the uniform

team selects the best alternative, namely $\sigma_s^*(1)$. Finally, agent $i$ is chosen to maximize the overall accuracy $\mathbb{E}[X_{is}^k]$, where the expectation is over the state $s$ and the draws from $P_i^1$ and $P_i^2$.

The diverse team consists of one copy of each agent $i \in N$. Importantly, although we can take multiple copies of each agent and a total of $k$ copies, we show that taking even a single copy of each agent outperforms the uniform team. Each agent $i$ has its own biased truth $\sigma_{is}$ drawn from $P_i^1(\boldsymbol{\theta}_s)$, and it draws its vote $\psi_{is}$ from $P_i^2(\sigma_{is})$. This results in the profile $\boldsymbol{\psi}_s^n = (\psi_{1s}, \ldots, \psi_{ns})$. Let $Y_s^n = \mathbb{I}[f(\boldsymbol{\psi}_s^n) = \sigma_s^*(1)]$ be the indicator random variable denoting whether the diverse team selects the best alternative, namely $\sigma_s^*(1)$.

Below we put forward a number of assumptions on noise models; different subsets of assumptions are required for different results. We remark that each agent $i \in N$ has two noise models *for each possible number of alternatives* $m$. However, for the sake of notational convenience, we refer to these noise models as $P_i^1$ and $P_i^2$ irrespective of $m$. This is natural, as the classic noise models defined in Section 2.1 describe a noise model for each $m$.

**A1** For each agent $i \in N$, the associated noise models $P_i^1$ and $P_i^2$ are neutral.

**A2** There exists a universal constant $\eta > 0$ such that for each agent $i \in N$, every possible ground truth $\boldsymbol{\theta}$ (and the corresponding true ranking $\sigma^*$), and every $k \in [m]$ (where $m$ is the number of alternatives), $\Pr_{\sigma \sim P_i^1(\boldsymbol{\theta})}[\sigma^*(1) = \sigma(k)] \le 1 - \eta$.

In words, assumption A2 requires that the true best alternative appear in any particular position with probability at most a constant which is less than 1. This ensures that the noise model indeed introduces a non-zero constant amount of noise in the position of the true best alternative.

**A3** There exists a universal constant $\alpha > 0$ such that for each agent $i \in N$, the noise models $P_i^1$ and $P_i^2$ are PD-$\alpha$.

**A4** There exists a universal constant $\alpha > 0$ such that for each agent $i \in N$, the noise models $P_i^1$ and $P_i^2$ are PPM-$\alpha$ and PM-$\alpha$, respectively.

We show that the preceding assumptions are indeed very mild in that classical noise models introduced in Section 2.1 satisfy all four assumptions. The proof of the following result appears in Appendix E.

**Theorem 1.** *With a fixed set of alternatives (such that the true qualities of every two alternatives are distinct in the case where the ground truth is the set of true qualities), the Mallows-$\phi$ model with $\phi \in [\rho, 1 - \rho]$, the Thurstone-Mosteller model with variance parameter $\sigma^2 \in [L, U]$, and the Plackett-Luce model all satisfy assumption A1, A2, A3, and A4, given that $\rho \in (0, 1/2)$, $L > 0$, and $U > L$ are constants.*

We are now ready to present our main result; its proof appears in Appendix F.

**Theorem 2.** *Let $\mu$ be a distribution over the state space $S$. Let the set of alternatives in all states $\{A_s\}_{s \in S}$ be fixed.*

1. *Under the assumptions A1 and A2, and for any neutral voting rule $f$, there exists a universal constant $c > 0$ such that for every $k$ and every $N = \{1, \ldots, n\}$, it holds that $\max_{i \in N} \mathbb{E}[X_{is}^k] \le 1 - c$, where the expectation is over the state $s \sim \mu$, the ground truths $\sigma_{is} \sim P_i^1(\boldsymbol{\theta}_s)$ for all $s \in S$, and the votes $\pi_{is}^j \sim P_i^2(\sigma_{is})$ for all $j \in [k]$.*

2. *Under each of the following two conditions, for a voting rule $f$, it holds that $\lim_{n \to \infty} \mathbb{E}[Y_s^n] = 1$, where the expectation is over the state $s \sim \mu$, the biased truths $\sigma_{is} \sim P_i^1(\boldsymbol{\theta}_s)$ for all $i \in N$ and $s \in S$, and the votes $\psi_{is} \sim P_i^2(\sigma_{is})$ for all $i \in N$ and $s \in S$: (i) assumptions A1 and A3 hold, and $f$ is PD-c; (ii) assumptions A1 and A4 hold, and $f$ is Condorcet consistent.*

## 4 Experimental Results

We now present our experimental results in the Computer Go domain. We use a novel methodology for generating large teams, which we view as one of our main contributions. It is fundamentally

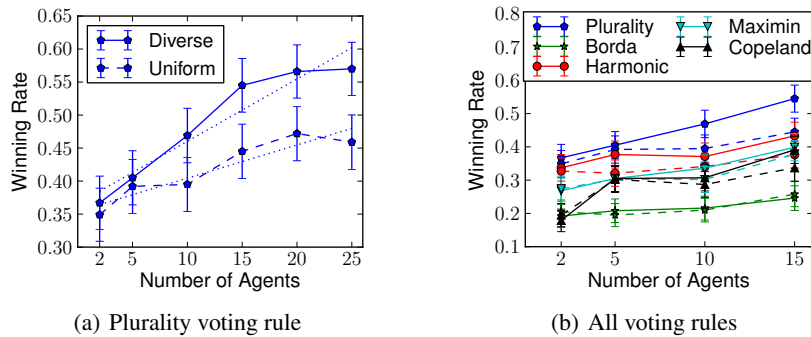

Figure 1: Winning rates for Diverse (continuous line) and Uniform (dashed line), for a variety of team sizes and voting rules.

different from that of Marcolino et al. [19, 20], who created a diverse team by combining four different, independently developed Go programs. Here we automatically create arbitrarily many diverse agents by parameterizing one Go program. Specifically, we use different parametrizations of Fuego 1.1 [10]. Fuego is a state-of-the-art, open source, publicly available Go program; it won first place in $19 \times 19$ Go in the Fourth Computer Go UEC Cup, 2010, and also won first place in $9 \times 9$ Go in the 14th Computer Olympiad, 2009. We sample random values for a set of parameters for each generated agent, in order to change its behavior. In Appendix G we list the sampled parameters, and the range of sampled values. The original Fuego is the strongest agent, as we show in Appendix H.

All results were obtained by simulating 1000 $9 \times 9$ Go games, in an HP dl165 with dual dodeca core, 2.33GHz processors and 48GB of RAM. We compare the winning rates of games played against a fixed opponent. In all games the system under evaluation plays as white, against the original Fuego playing as black. We evaluate two types of teams: *Diverse* is composed of different agents, and *Uniform* is composed of copies of a specific agent (with different random seeds). In order to study the performance of the uniform team, for each sample (which is an entire Go game) we construct a team consisting of copies of a randomly chosen agent from the diverse team. Hence, the results presented for Uniform are approximately the mean behavior of all possible uniform teams, given the set of agents in the diverse team. In all graphs, the error bars show 99% confidence intervals.

Fuego (and, in general, all programs using Monte Carlo tree search algorithms) is not originally designed to output a ranking over all possible moves (alternatives), but rather to output a single move — the best one according to its search tree (of course, there is no guarantee that the selected move is in fact the best one). In this paper, however, we wish to compare plurality (which only requires each agent's top choice) with voting rules that require an entire ranking from each agent. Hence, we modified Fuego to make it output a ranking over moves, by using the data available in its search tree (we rank by the number of simulations per alternative). We ran games under 5 different voting rules: plurality, Borda count, the harmonic rule, maximin, and Copeland. Plurality, Borda count (which we limit to the top 6 positions in the rankings), and the harmonic rule (see Appendix A) are PD-c rules, while maximin and Copeland are Condorcet-consistent rules (see, e.g., [24]).

We first discuss Figure 1(a), which shows the winning rates of Diverse and Uniform for a varying number of agents using the plurality voting rule. The winning rates of both teams increase as the number of agents increases. Diverse and Uniform start with similar winning rates, around 35% with 2 agents and 40% with 5 agents, but with 25 agents Diverse reaches 57%, while Uniform only reaches 45.9%. The improvement of Diverse over Uniform is not statistically significant with 5 agents ($p = 0.5836$), but is highly statistically significant with 25 agents ($p = 8.592 \times 10^{-7}$). We perform linear regression on the winning rates of the two teams to compare their rates of improvement in performance as the number of agents increases. Linear regression (shown as the dotted lines in Figure 1(a)) gives the function $y = 0.0094x + 0.3656$ for Diverse ($R^2 = 0.9206$, $p = 0.0024$) and $y = 0.0050x + 0.3542$ for Uniform ($R^2 = 0.8712$, $p = 0.0065$). In particular, the linear approximation for the winning rate of Diverse increases roughly twice as fast as the one for Uniform as the number of agents increases.

Despite the strong performance of Diverse (it beats the original Fuego more than 50% of the time), it seems surprising that its winning rate converges to a constant that is significantly smaller than 1, in light of Theorem 2. There are (at least) two reasons for this apparent discrepancy. First, Theorem 2 deals with the probability of making good moves in individual board positions (states), whereas the figure shows winning rates. Even if the former probability is very high, a bad decision in a single state of a game can cost Diverse the entire game. Second, our diverse team is formed by randomly sampling different parametrizations of Fuego. Hence, there might still exist a subset of world states where all agents would play badly, regardless of the parametrization. In other words, the parametrization procedure may not be generating the idealized diverse team (see Appendix H).

Figure 1(b) compares the results across different voting rules. As mentioned above, to generate ranked votes, we use the internal data in the search tree of an agent's run (in particular, we rank using the number of simulations per alternative). We can see that increasing the number of agents has a positive impact for all voting rules under consideration. Moving from 5 to 15 agents for Diverse, plurality has a 14% increase in the winning rate, whereas other voting rules have a mean increase of only 6.85% ($std = 2.25\%$), close to half the improvement of plurality. For Uniform, the impact of increasing the number of agents is much smaller: Moving from 5 to 15 agents, the increase for plurality is 5.3%, while the mean increase for other voting rules is 5.70% ($std = 1.45\%$). Plurality surprisingly seems to be the best voting rule in these experiments, even though it uses less information from the submitted rankings. This suggests that the ranking method used does not typically place good alternatives in high positions other than the very top.

Hence, we introduce a novel procedure to generate rankings, which we view as another major methodological contribution. To generate a ranked vote from an agent on a given board state, we run the agent on the board state 10 times (each run is independent of other runs), and rank the moves by the number of times they are played by the agent. We use these votes to compare plurality with the four other voting rules, for Diverse with 5 agents. Figure 2 shows the results. All voting rules outperform plurality; Borda and maximin are sta-

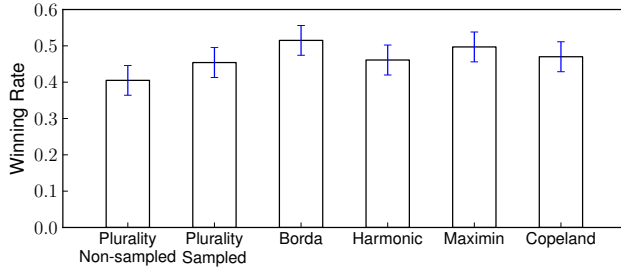

Figure 2: All voting rules, for Diverse with 5 agents, using the new ranking methodology.

tistically significantly better ($p < 0.007$ and $p = 0.06$, respectively). All ranked voting rules are also statistically significantly better than the non-sampled (single run) version of plurality.

## 5 Discussion

While we have focused on computer Go for motivation, we have argued in Section 1 that our theoretical model is more widely applicable. At the very least, it is relevant to modeling game-playing agents in the context of other games. For example, random sampling techniques play a key role in the design of computer poker programs [25]. A complication in some poker games is that the space of possible moves, in some stages of the game, is infinite, but this issue can likely be circumvented via an appropriate discretization.

Our theoretical model does have (at least) one major shortcoming when applied to multistage games like Go or poker: it assumes that the state space is "flat". So, for example, making an excellent move in one state is useless if the agent makes a horrible move in a subsequent state. Moreover, rather than having a fixed probability distribution $\mu$ over states, the agents' strategies actually determine which states are more likely to be reached. To the best of our knowledge, existing models of voting do not capture sequential decision making — possibly with a few exceptions that are not relevant to our setting, such as the work of Parkes and Procaccia [22]. From a theoretical and conceptual viewpoint, the main open challenge is to extend our model to explicitly deal with sequentiality.

**Acknowledgments:** Procaccia and Shah were partially supported by the NSF under grants IIS-1350598 and CCF-1215883, and Marcolino by MURI grant W911NF-11-1-0332.

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
