[Supplementary Material]

## A  Voting Rules

We first define positional scoring rules and Bucklin's rule, and then prove Proposition 1.

**Positional Scoring Rules**  A positional scoring rule is given by a scoring vector $(\alpha_1, \ldots, \alpha_m)$ where $\alpha_i \geq \alpha_{i+1}$ for all $i \in \{1, \ldots, m\}$ and $\alpha_1 > \alpha_m$. Under this rule for each vote $\sigma$ and $i \in [m]$, $\alpha_i$ points are awarded to the alternative $\sigma(i)$. The alternative with the most points overall is selected as the winner. The proof of Proposition 1 holds irrespective of the tie-breaking rule used. Special positional scoring rules include *plurality* with scoring vector $(1, 0, 0, \ldots, 0)$, *Borda count* with scoring vector $(m, m-1, \ldots, 1)$, the *veto* rule with scoring vector $(1, 1, \ldots, 1, 0)$, and the *harmonic* rule [5] with scoring vector $(1, 1/2, \ldots, 1/m)$.

**Bucklin's rule**  The Bucklin score of an alternative $a$ is the minimum $k$ such that $a$ is among the first $k$ positions in the majority of input votes. Bucklin's rule outputs the alternative with the lowest Bucklin score, and breaks ties among alternatives with the same Bucklin score $\ell$ according to the number of rankings that have the alternative in the first $\ell$ positions.

**Proposition 1.** *All positional scoring rules (including plurality and Borda count) and Bucklin's rule are PD-c rules.*

*Proof.* Consider a profile $\pi$ with $n$ rankings and a position-dominating winner $a$. We show that any positional scoring rule as well as Bucklin's rule outputs $a$ on $\pi$. For any $j \in \{1, \ldots, m-1\}$, let $T_\pi(c, j)$ denote the number of votes where alternative $c$ is among the first $j$ positions in $\pi$.

For Bucklin's rule, consider arbitrary alternative $a' \neq a$. Let $k$ denote the Bucklin score of $a$ and $k'$ denote the Bucklin score of $a'$. If $k > k'$, then $T_\pi(a', k') > n/2$ and $T_\pi(a, k') \leq T_\pi(a, k-1) < n/2$, which is impossible since the $a$ is the position-dominating winner in $\pi$. If $k < k'$, then Bucklin's rule would select $a$ over $a'$, as required.

If $k = k' \neq m$, then we have $T_\pi(a, k) > T_\pi(a', k)$ because $a$ is the position-dominating winner. Hence, the tie is broken in favor of $a$. Lastly, we note that $k = k' = m$ is not possible because it would imply that the total number of appearances of $a$ and $a'$ in the last position is $n - T_\pi(a, m-1) + n - T_\pi(a', m-1) > 2 \cdot n - 2 \cdot T_\pi(a, m-1) > n$. Thus, Bucklin's rule would choose $a$ over every other alternative $a'$, i.e., it would output $a$ as the winner, as required.

Consider a positional scoring rule with scoring vector $(\alpha_1, \ldots, \alpha_m)$. As shown in the proof of Theorem 3.10 in [8], the score of an alternative $a'$ in $\pi$ is equivalently given by $\sum_{k=1}^{m-1} \beta_k \cdot T_\pi(a', k)$, where $\beta_i = \alpha_i - \alpha_{i-1} \geq 0$. It is now easy to see that the position-dominating winner $a$ would have strictly higher score than every other alternative because $\beta_i > 0$ for some $i$. Hence, every positional scoring rule would also output $a$, as required. $\qquad\square$

## B  Proof of Lemma 1

Let $m$ be the number of alternatives. The result is trivial for $m = 1$. For $m \geq 2$, let $P^1$ and $P^2$ be PD-$\alpha_1$ and PD-$\alpha_2$ noise models, respectively. For $i, j \in [m]$, let $T^1(i, j)$ and $T^2(i, j)$ be the probabilities that the $i^{th}$ alternative in the true ranking is placed in position $j$ in a sample from $P^1$ and $P^2$, respectively.[1] For $i, j \in [m]$, let $F^1(i, j) = \sum_{l=1}^j T^1(i, l)$ and $F^2(i, j) = \sum_{l=1}^j T^2(i, l)$. Now, fix $1 \leq p < q \leq m$. The difference between the probabilities of the $p^{th}$ and $q^{th}$ alternatives in the true ranking appearing among the first $k$ positions in a sample from $P^2 \circ P^1$ is

$$\sum_{i=1}^k \sum_{j=1}^m T^1(p, j) T^2(j, i) - \sum_{i=1}^k \sum_{j=1}^m T^1(q, j) T^2(j, i)$$

$$= \sum_{i=1}^k \sum_{j=1}^m T^2(j, i) \left( T^1(p, j) - T^1(q, j) \right) = \sum_{j=1}^m F^2(j, k) \left( T^1(p, j) - T^1(q, j) \right)$$

$$= \sum_{j=1}^m \left( F^2(j, k) - F^2(j+1, k) \right) \left( F^1(p, j) - F^1(q, j) \right) \geq \alpha_1 \cdot \alpha_2,$$

where the second transition follows by interchanging the order of summation, the third transition follows by simple algebra (we let $F^2(m+1, k) = 0$ because there are only $m$ alternatives), and the last transition holds because for $j = 1$, the two terms in the summation are at least $\alpha_2$ and $\alpha_1$, respectively, as $P^1$ and $P^2$ are PD-$\alpha_1$ and PD-$\alpha_2$ noise models, respectively, and for $j > 1$, both terms are non-negative. □

## C   Example: Composition of PM Noise Models

Consider a neutral noise model $P$ over 3 alternatives that is of the rank-to-rank type. We describe the probabilities of various rankings when the ground truth is $a \succ b \succ c$ (the probabilities of various permutations of the true ranking are independent of the true ranking): $\Pr_P[a \succ b \succ c; a \succ b \succ c] = 0.51$, $\Pr_P[b \succ a \succ c; a \succ b \succ c] = 0.09$, $\Pr_P[c \succ b \succ a; a \succ b \succ c] = 0.4$. That is, the true ranking stays unchanged with probability $0.51$, the top two alternative are swapped with probability $0.09$, and the first and the last alternatives are swapped with probability $0.4$.

Clearly, $P$ is PM-$0.02$ because every pairwise comparison is preserved with probability at least $0.51$. Let us consider the composition $P \circ P$. We evaluate the probability that the top alternative in the true ranking stays above the second alternative in the true ranking in a vote sampled from $P \circ P$. This probability is precisely

$$0.51 \cdot 0.51 + 0.09 \cdot 0.49 + 0.4 \cdot 0.4 = 0.4642 < 0.5.$$

Hence, in $P \circ P$ the pair of top two alternatives in the true ranking is flipped with probability more than $0.5$. It follows that $P \circ P$ is not PM-$\alpha$ for any $\alpha > 0$.

## D   Proof of Lemma 2

The result is trivial for $m = 1$. Let $m \geq 1$. Let $P^1$ and $P^2$ be PPM-$\alpha_1$ and PM-$\alpha_2$ noise models respectively. Let $m$ denote the number of alternatives. Fix $1 \leq i < j \leq m$. For $1 \leq i' < j' \leq m$, define $T^1(i, j, i', j')$ to be the probability that the alternatives in positions $i$ and $j$ in the true ranking appear in positions $i'$ and $j'$, respectively, in a vote sampled from $P^1$. Since $P^1$ is PPM-$\alpha_1$, we have $T^1(i, j, i', j') \geq T^1(i, j, j', i')$ for all $1 \leq i' < j' \leq m$ and $T^1(i, j, i', j') \geq T^1(i, j, j', i') + \alpha_1$ for some $1 \leq i' < j \leq m'$.

Let $T^2(i, j)$ denote the probability that the alternative in position $i$ in the true ranking is preferred to the alternative in position $j$ in the true ranking in a vote sampled from $P^2$. Since $P^2$ is PM-$\alpha_2$, we know that $T^2(i, j) \geq T^2(j, i) + \alpha_2$. Now, the difference between the probability of the alternative in position $i$ in the true ranking being preferred to the alternative in position $j$ in the true ranking in a vote from $P^2 \circ P^1$ and the probability of its converse is

$$\sum_{i', j' \in [m]} T^1(i, j, i', j') \cdot T^2(i', j') - \sum_{i', j' \in [m]} T^1(i, j, i', j') \cdot T^2(j', i')$$

$$= \sum_{i', j' \in [m]} T^1(i, j, i', j') \cdot \left(T^2(i', j') - T^2(j', i')\right)$$

$$= \sum_{1 \leq i' < j' \leq m} \left(T^1(i, j, i', j') \cdot \left(T^2(i', j') - T^2(j', i')\right) + T^1(i, j, j', i') \cdot \left(T^2(j', i') - T^2(i', j')\right)\right)$$

$$= \sum_{1 \leq i' < j' \leq m} \left(T^1(i, j, i', j') - T^1(i, j, j', i')\right) \cdot \left(T^2(i', j') - T^2(j', i')\right) \geq \alpha_1 \cdot \alpha_2,$$

where the last transition holds because our assumptions on $P^1$ and $P^2$ imply that there exist $1 \leq i' < j' \leq m$ for which $T^1(i, j, i', j') \geq T^1(i, j, j', i') + \alpha_1$, and for those values of $i'$ and $j'$, we have $T^2(i', j') \geq T^2(j', i') + \alpha_2$. Thus, $P^2 \circ P^1$ is PM-$(\alpha_1 \cdot \alpha_2)$. □

## E   Proof of Theorem 1

We show that three classical noise models—the Mallows-$\phi$ model, the Thurstone-Mosteller model, and the Plackett-Luce model—satisfy our four assumptions. For assumption A4, we only show that the noise models are PPM-$\alpha$; this implies that they are also PM-$\alpha$.

*Proof of Theorem 1.* Let $m$ denote the number of alternatives. Let $\mathcal{L}(A)$ be the set of all rankings over $m$ alternatives. In all the proofs below, $P$ will denote the noise model under consideration, and $p_{i,j}$ will denote the probability that the alternative in position $i$ in the true ranking appears in position $j$ in a vote sampled from $P$. We begin with the proof for the Mallows-$\phi$ model.

**Part I: The Mallows-$\phi$ model.**

We prove that the Mallows-$\phi$ model with $\phi \in [\rho, 1 - \rho]$ and constant $\rho \in (0, 1/2)$ satisfies assumptions A1, A2, A3, and A4.

**Assumption A1:** It is well-known and easy to check that neutrality of the Kendall-Tau distance implies neutrality of the Mallows-$\phi$ model for all $\phi \in (0, 1)$. Hence, the Mallows-$\phi$ model satisfies assumption A1.

**Assumption A2:** We need to show that there exists a constant $\eta > 0$ such that $p_{1,k} \leq 1 - \eta$ for all $k \in [m]$. Lemma 3.8 in [8] shows that when $\sigma$ is sampled from the Mallows-$\phi$ model with true ranking $\sigma^*$ and $m$ alternatives,

$$p_{i,1} = \frac{\phi^{i-1}}{\sum_{j=0}^{m-1} \phi^j}.$$

The proof explicitly evaluates the probability by summing the probabilities of all rankings where $\sigma(1) = \sigma^*(i)$. Using an almost identical proof technique, we evaluate the similar probability $p_{1,i}$ used in assumption A2. First, we show that for any $i \in [m-1]$, we have $p_{1,i+1} = \phi \cdot p_{1,i}$. To see this,

$$
\begin{aligned}
p_{1,i} - p_{1,i+1} &= \frac{\sum_{\sigma \in \mathcal{L}(A) | \sigma(i) = \sigma^*(1)} \phi^{d_{KT}(\sigma,\sigma^*)} - \sum_{\sigma \in \mathcal{L}(A) | \sigma(i+1) = \sigma^*(1)} \phi^{d_{KT}(\sigma,\sigma^*)}}{Z_\phi^m} \\
&= \frac{\sum_{\sigma \in \mathcal{L}(A) | \sigma(i) = \sigma^*(1)} \left( \phi^{d_{KT}(\sigma,\sigma^*)} - \phi^{d_{KT}(\sigma_{\sigma(i) \leftrightarrow \sigma(i+1)}, \sigma^*)} \right)}{Z_\phi^m} \\
&= \sum_{\sigma \in \mathcal{L}(A) | \sigma(i) = \sigma^*(1)} \frac{\phi^{d_{KT}(\sigma,\sigma^*)} \cdot (1 - \phi)}{Z_\phi^m} = (1 - \phi) \cdot p_{1,i},
\end{aligned}
$$

where the second transition holds because $\sigma \leftrightarrow \sigma_{\sigma(i) \leftrightarrow \sigma(i+1)}$ is a bijection between the set of rankings where $\sigma^*(1) = \sigma(i)$ and the set of rankings where $\sigma^*(1) = \sigma(i+1)$. The third transition holds because swapping $\sigma(i) = \sigma^*(1)$ with the alternative $\sigma(i+1)$ does not change any pairwise comparisons between $\sigma$ and $\sigma^*$, except that of $\sigma^*(1)$ and $\sigma(i+1)$. The latter is mismatched with $\sigma^*$ after the exchange. Hence, the Kendall Tau distance to $\sigma^*$, which is equal to the number of pairwise mismatches with $\sigma^*$, increases by exactly 1 after the exchange.

Hence, $p_{1,i} - p_{1,i+1} = (1 - \phi) \cdot p_{1,i}$, which implies that $p_{1,i+1} = \phi \cdot p_{1,i}$. Applying this repeatedly, we have that $p_{1,i} = p_{1,1} \cdot \phi^{i-1}$, for every $i \in [m]$. Summing over $i \in [m]$ and observing that $\sum_{i=1}^m p_{1,i} = 1$, we get that

$$p_{1,i} = \frac{\phi^{i-1}}{\sum_{j=0}^{m-1} \phi^j} \leq \frac{\phi^{i-1}}{\sum_{j=0}^{\infty} \phi^j} = \phi^{i-1} \cdot (1 - \phi).$$

Hence, for all $i \in [m]$, $p_{1,i} \leq p_{1,1} \leq 1 - \phi \leq 1 - \rho$. Hence, the Mallows-$\phi$ model satisfies assumption A2 with $\eta = \rho$.

**Assumption A3:** We need to show that for all $i, j \in [m]$ with $i < j$ and $k \in [m-1]$,

$$\Pr_P[\sigma^*(i) \in \sigma([k])] > \Pr_P[\sigma^*(j) \in \sigma([k])]. \tag{5}$$

We take the difference of the two terms, and remove the set of common rankings where both $\sigma^*(i)$ and $\sigma^*(j)$ are in $\sigma([k])$. Thus, we get

$$
\Pr_P[\sigma^*(i) \in \sigma([k])] - \Pr_P[\sigma^*(j) \in \sigma([k])] = \sum_{\substack{\sigma \in \mathcal{L}(A) \\ \sigma^*(i) \in \sigma([k]) \\ \sigma^*(j) \notin \sigma([k])}} \Pr_P[\sigma; \sigma^*] - \sum_{\substack{\sigma \in \mathcal{L}(A) \\ \sigma^*(j) \in \sigma([k]) \\ \sigma^*(i) \notin \sigma([k])}} \Pr_P[\sigma; \sigma^*]
$$

$$
= \sum_{\substack{\sigma \in \mathcal{L}(A) \\ \sigma^*(i) \in \sigma([k]) \\ \sigma^*(j) \notin \sigma([k])}} \left( \Pr_P[\sigma; \sigma^*] - \Pr_P[\sigma_{\sigma^*(i) \leftrightarrow \sigma^*(j)}; \sigma^*] \right) \quad (6)
$$

$$
\geq (1 - \phi) \cdot \sum_{\substack{\sigma \in \mathcal{L}(A) \\ \sigma^*(i) \in \sigma([k]) \\ \sigma^*(j) \notin \sigma([k])}} \Pr_P[\sigma; \sigma^*]
$$

$$
\geq (1 - \phi) \cdot \frac{\phi^{-m^2}}{Z_\phi^m} \geq \rho^{1-m-m^2},
$$

where the second transition holds because $\sigma \leftrightarrow \sigma_{\sigma^*(i) \leftrightarrow \sigma^*(j)}$ is a bijection between the set of rankings where $\sigma^*(i) \in \sigma([k])$ and $\sigma^*(j) \notin \sigma([k])$, and the set of rankings where $\sigma^*(i) \notin \sigma([k])$ and $\sigma^*(j) \in \sigma([k])$. The third transition holds because $\sigma$ matches with $\sigma^*$ in the pairwise comparison of $\sigma^*(i)$ and $\sigma^*(j)$ (because $\sigma^*(i) \in \sigma([k])$ and $\sigma^*(j) \notin \sigma([k])$), thus swapping them increases its distance from $\sigma^*$ by at least 1 due to the swap-increasing property of the Kendall-Tau distance (Lemma 3.5 in [8]). Hence, the probability drops at least by a factor of $\phi$ from Equation (1). The fourth transition holds because there is at least one ranking where $\sigma^*(i) \in \sigma([k])$ and $\sigma^*(j) \notin \sigma([k])$, and the ranking has probability at least $\phi^{-m^2}/Z_\phi^m$. Finally, the last transition holds because $1/Z_\phi^m \geq (1 - \phi)^m$ (which is easy to show), and $\phi \in [\rho, 1 - \rho]$.

Hence, there exists a constant $\alpha = \rho^{1-m-m^2} > 0$ such that $\Pr[\sigma^*(i) \in \sigma([k])] \geq \Pr[\sigma^*(j) \in \sigma([k])] + \alpha$ for all $i, j \in [m]$ with $i < j$ and $k \in [m-1]$, as desired.

**Assumption A4:** We want to show that there exists a constant $\alpha > 0$ such that for all $i, j \in [m]$ with $i < j$, the quantity

$$
\Pr_P[\sigma^*(i) = \sigma(i') \wedge \sigma^*(j) = \sigma(j')] - \Pr_P[\sigma^*(i) = \sigma(j') \wedge \sigma^*(j) = \sigma(i')] \quad (7)
$$

is non-negative for all $i', j' \in [m]$ with $i' < j'$ and at least $\alpha$ for some $i', j' \in [m]$ with $i' < j'$.

Fix $i, j, i', j' \in [m]$ such that $i < j$ and $i' < j'$. Similarly to Equation (6), we note that $\sigma \leftrightarrow \sigma_{\sigma^*(i) \leftrightarrow \sigma^*(j)}$ is also a bijection between the set of rankings where $\sigma^*(i) = \sigma(i') \wedge \sigma^*(j) = \sigma(j')$, and the set of rankings where $\sigma^*(i) = \sigma(j') \wedge \sigma^*(j) = \sigma(i')$. Hence, following the same steps, we can derive

$$
\Pr_P[\sigma^*(i) = \sigma(i') \wedge \sigma^*(j) = \sigma(j')] - \Pr_P[\sigma^*(i) = \sigma(j') \wedge \sigma^*(j) = \sigma(i')]
$$
$$
\geq (1 - \phi) \cdot \Pr_P[\sigma^*(i) = \sigma(i') \wedge \sigma^*(j) = \sigma(j')].
$$

Thus, the difference is always non-negative. Further, note that there always exists a ranking $\sigma$ where $\sigma^*(i) = \sigma(i') \wedge \sigma^*(j) = \sigma(j')$. Thus, using the same bound as in the case of assumption A3, we get that there exists a constant $\alpha > 0$ depending only on $\rho$ and $m$ such that the quantity in Equation (7) is at least $\alpha$ for all $1 \leq i' < j' \leq m$.

**PART II: The Thurstone-Mosteller and the Plackett-Luce models.**

We give a common proof by viewing both noise models as special cases of a random utility model. Let $\theta_a$ denote the true quality of alternative $a$. Let $\mu_a(\theta_a)$ denote the distribution from which noisy estimate of the quality of alternative $a$ is sampled. Let $f_{\theta_a}$ and $F_{\theta_a}$ denote the PDF and CDF, respectively, of $\mu_a(\theta_a)$, i.e., the noisy quality estimate comes from a distribution that only depends on the true quality. We assume the following two properties on $f_\theta$.

(P1) $f_\theta$ shifts with $\theta$, i.e., $f_\theta(x) = f_{\theta'}(x + \theta' - \theta)$ for all $x, \theta, \theta' \in \mathbb{R}$.

(P2) It is more likely that a higher quality estimate emerged from higher true quality and lower quality estimate emerged from lower true quality, than vice-versa. Formally, for all $\theta > \theta'$ and $x > x'$, $f_\theta(x) \cdot f_{\theta'}(x') - f_\theta(x') \cdot f_{\theta'}(x) > 0$. Further, for $x$, $x'$, $\theta$, $\theta'$, $x - x'$, and $\theta - \theta'$ bounded from above and below by constants the difference is at least a constant.

(P3) A random variable $X$ distributed according to $f_\theta$ satisfies that $\Pr[|X - \theta| \leq c]$, $\Pr[X > \theta + c]$, and $\Pr[X < \theta - c]$ are all positive constants less than 1 if $c$ is a positive constant. Further, the set of values $\{f_\theta(\theta + x)\}_{x \in [-c,c]}$ is bounded from above and below by positive constants if $c$ is a constant.

**Lemma 3.** *The normal distribution (with variance parameter bounded from above and below by positive constants) and the Gumbel distribution satisfy properties P1, P2, and P3.*

*Proof.* Property P1 follows directly from the definition of the two distributions. For property P2, first consider the normal distribution with fixed variance $\nu$.

$$f_\theta(x) \cdot f_{\theta'}(x') - f_\theta(x') \cdot f_{\theta'}(x)$$
$$= \frac{1}{2\pi\nu^2} \left( e^{-\frac{(x-\theta)^2}{2\nu^2}} \cdot e^{-\frac{(x'-\theta')^2}{2\nu^2}} - e^{-\frac{(x'-\theta)^2}{2\nu^2}} \cdot e^{-\frac{(x-\theta')^2}{2\nu^2}} \right)$$
$$= \frac{1}{2\pi\nu^2} \left( e^{-\frac{(x-\theta)^2 + (x'-\theta')^2}{2\nu^2}} - e^{-\frac{(x'-\theta)^2 + (x-\theta')^2}{2\nu^2}} \right) > 0,$$

where the last transition holds because

$$(x' - \theta)^2 + (x - \theta')^2 - (x - \theta)^2 - (x' - \theta')^2$$
$$= 2 \left( x\theta + x'\theta' - x\theta' - x'\theta \right)$$
$$= 2(x - x')(\theta - \theta') > 0.$$

Similarly, for the Gumbel distribution, we have

$$f_\theta(x) \cdot f_{\theta'}(x') - f_\theta(x') \cdot f_{\theta'}(x)$$
$$= e^{-(x-\theta) - e^{-(x-\theta)}} \cdot e^{-(x'-\theta') - e^{-(x'-\theta')}} - e^{-(x'-\theta) - e^{-(x'-\theta)}} \cdot e^{-(x-\theta') - e^{-(x-\theta')}}$$
$$= e^{-x+\theta-x'+\theta'} \cdot \left( e^{-e^{-(x-\theta)} - e^{-(x'-\theta')}} - e^{-e^{-(x'-\theta)} - e^{-(x-\theta')}} \right).$$

Finally, we have that

$$e^{-(x-\theta)} + e^{-(x'-\theta')} - e^{-(x'-\theta)} - e^{-(x-\theta')}$$
$$= \left( e^{-x'} - e^{-x} \right) \left( e^{\theta'} - e^{\theta} \right) < 0.$$

In both of these cases, it can easily be checked that the difference is at least a positive constant if $x$, $x'$, $\theta$, $\theta'$, $x - x'$, and $\theta - \theta'$ are bounded from both sides by constants.

For property P3, this is a well-known fact for the normal distribution when the standard deviation $\sigma$ itself is bounded from above and below by constants. For the Gumbel distribution, this can be checked using its explicit PDF $f_\theta(x) = e^{-(x-\theta) - e^{-(x-\theta)}}$ and its explicit CDF $F_\theta(x) = e^{-e^{-(x-\theta)}}$.

Hence, both distributions satisfy all three properties. $\qquad\square$

Next, we show that any random utility model where the PDF satisfies these three properties satisfies our four assumptions. Recall that the set of alternatives and therefore their true qualities are fixed. We use a slightly different notation for the following proofs. For $i \in [m]$, let the true quality of alternative $\sigma^*(i)$ be $\theta_i$, and let its noisy quality estimate be the random variable $X_i$ whose value is drawn from $\mu_a(\theta_i)$, where $a = \sigma^*(i)$. Let $f_{X_i}$ and $F_{X_i}$ denote the PDF and CDF of $X_i$ respectively. Let $X_{-i}$ denote the set of random variables $\{X_1, \ldots, X_{i-1}, X_{i+1}, \ldots, X_m\}$.

For a set $S$ of $t$ random variables and $k \in [t]$, let $\text{top}_k(S)$ be the random variable denoting the $k^{th}$ highest value among the random variables in $S$, i.e., it is the $t - k + 1^{th}$ order statistic of $S$. Correspondingly, let $f_{\text{top}_k(S)}$ and $F_{\text{top}_k(S)}$ denote its CDF and PDF respectively.

**Assumption A1:** Neutrality is evident because the noisy quality estimates (and therefore the ranking of the alternatives in a sample vote) depend on the true qualities of the alternatives, but are independent of the identities of the alternatives.

**Assumption A2:** We want to show that there exists a constant $\eta > 0$ such that $p_{1,i} \leq 1 - \eta$ for all $i \in [m]$. Note that

$$p_{1,1} \geq \Pr\left[X_1 > \frac{\theta_1 + \theta_2}{2}\right] \cdot \prod_{i=2}^{m} \Pr\left[X_i < \frac{\theta_1 + \theta_2}{2}\right].$$

On the right hand side, we use the fact that $X_i$ are independent of each other. Each term represents the probability of an $X_i$ bounded within a constant amount on one side. Hence, due to property (P3) and the fact that $m$ is constant, we have that $p_{1,1} \geq \eta_1$ for some constant $\eta_1 > 0$. This immediately implies that $p_{1,i} \leq 1 - \eta_1$ for all $i \in \{2, \ldots, m\}$. We now prove that $p_{1,1} \leq 1 - \eta_2$ for some constant $\eta_2 > 0$. This is sufficient because it then follows that assumption A2 is satisfied with $\eta = \min(\eta_1, \eta_2) > 0$.

Note that

$$p_{1,1} \leq 1 - \Pr\left[X_1 < \frac{\theta_1 + \theta_2}{2}\right] \cdot \Pr\left[X_2 > \frac{\theta_1 + \theta_2}{2}\right].$$

Once again, both probability terms on the right hand side are positive constants due to property P3. Hence, $p_{1,1}$ is a constant less than 1. Hence, we have that assumption A2 is satisfied.

**Assumption A3:** We want to show that there exists a constant $\alpha > 0$ such that for all $i, j \in [m]$ with $i < j$ and $k \in [m-1]$, we have $\Pr_P[\sigma^*(i) \in \sigma([k])] \geq \Pr_P[\sigma^*(j) \in \sigma([k])] + \alpha$.

Note that $\sigma^*(i) \in \sigma([k])$ is the probability that the quality estimate of $\sigma^*(i)$ is among the $k$ highest quality estimates. This is equivalent to the quality estimate $X_i$ of $\sigma^*(i)$ being higher than the $k^{th}$ highest quality estimate among $X_{-i}$. Hence,

$$\Pr_P[\sigma^*(i) \in \sigma([k])] = \int_{t=-\infty}^{\infty} f_{X_i}(t) F_{\text{top}_k(X_{-i})}(t)\, \mathrm{d}t. \tag{8}$$

Let $\Delta = \theta_i - \theta_j$. Now, similarly to Equation (8) we also have

$$\Pr_P[\sigma^*(j) \in \sigma([k])] = \int_{t=-\infty}^{\infty} f_{X_j}(t) F_{\text{top}_k(X_{-j})}(t)\, \mathrm{d}t$$

$$= \int_{t=-\infty}^{\infty} f_{X_j}(t - \Delta) F_{\text{top}_k(X_{-j})}(t - \Delta)\, \mathrm{d}t. \tag{9}$$

Due to our assumption P1, we have $f_{X_i}(t) = f_{X_j}(t - \Delta)$ for all $t \in \mathbb{R}$. Now, we also have that for all $t \in \mathbb{R}$,

$$F_{\text{top}_k(X_{-j})}(t) \leq F_{\text{top}_k(X_{-i})}(t). \tag{10}$$

To see this, note that $X_{-j}$ and $X_{-i}$ have identical components except the former has $X_i$ and the latter has $X_j$ in its place. Note that $X_i$ first-order stochastically dominates $X_j$ because of our assumption P1. Hence, it follows that every order statistic of $X_{-j}$ first-order stochastically dominate the corresponding order statistic of $X_{-i}$. In particular, $\text{top}_k(X_{-j})$ first-order stochastically dominates $\text{top}_k(X_{-i})$, which is exactly Equation (10).

Substituting Equation (10) in the difference of Equations (8) and (9), we get

$$\Pr_P[\sigma^*(i) \in \sigma([k])] - \Pr_P[\sigma^*(j) \in \sigma([k])]$$

$$\geq \int_{\theta_j}^{\theta_i} f_{X_i}(t) \cdot \left(F_{\text{top}_k(X_{-j})}(t) - F_{\text{top}_k(X_{-j})}(t - \Delta)\right) \mathrm{d}t$$

$$= \int_{\theta_j}^{\theta_i} f_{X_i}(t) \cdot \Pr[\text{top}_k(X_{-j}) \in (t - \Delta, t]]\, \mathrm{d}t.$$

Now, due to property P3 and because a continuous function achieves its minimum on a closed interval, both $f_{X_i}(t)$ and $\Pr[\text{top}_k(X_{-j}) \in (t - \Delta, t]]$ are bounded from below by positive constants

for $t \in [\theta_j, \theta_i]$. Hence, the integral of their product is bounded from below by a positive constant, which implies that assumption A3 is satisfied.

**Assumption A4:** We show a stronger condition that there exists a positive constant $\alpha > 0$ such that for all $i, j, i', j' \in [m]$ with $i < j$ and $i' < j'$,

$$\Omega = \Pr{}_P[\sigma^*(i) = \sigma(i') \wedge \sigma^*(j) = \sigma(j')] - \Pr{}_P[\sigma^*(i) = \sigma(j') \wedge \sigma^*(j) = \sigma(i')] \geq \alpha. \quad (11)$$

Fix $i, j \in [m]$ with $i < j$ and $i', j' \in [m]$ with $i' < j'$. We explicitly evaluate the probability difference $\Omega$ in Equation (11) by conditioning on the values of $X_S = \{X_k\}_{k \in S}$, where $S = [m] \setminus \{i, j\}$. Consider two cases.

**Case 1:** $j' > i' + 1$ In this case, we want that $X_i \in (\text{top}_{i'}(X_S), \text{top}_{i'-1}(X_S)]$ and $X_j \in (\text{top}_{j'-1}(X_S), \text{top}_{j'-2}(X_S)]$. (The latter interval has shifted indices because insertion of $X_i$ would shift the rank of $X_j$ by one.) Thus, we can evaluate $\Delta$ in Equation (11) as follows.

$$\Omega = \int_{t_S} \int_{t_i = \text{top}_{i'}(X_S)}^{\text{top}_{i'-1}(X_S)} \int_{t_j = \text{top}_{j'-1}(X_S)}^{\text{top}_{j'-2}(X_S)} f_{X_i}(t_i) f_{X_j}(t_j) f_{X_S}(t_S) \, \mathrm{d}t_j \, \mathrm{d}t_i \, \mathrm{d}t_S$$

$$- \int_{t_S} \int_{t_j = \text{top}_{i'}(X_S)}^{\text{top}_{i'-1}(X_S)} \int_{t_i = \text{top}_{j'-1}(X_S)}^{\text{top}_{j'-2}(X_S)} f_{X_i}(t_i) f_{X_j}(t_j) f_{X_S}(t_S) \, \mathrm{d}t_j \, \mathrm{d}t_i \, \mathrm{d}t_S$$

$$= \int_{t_S} \int_{q = \text{top}_{i'}(X_S)}^{\text{top}_{i'-1}(X_S)} \int_{q' = \text{top}_{j'-1}(X_S)}^{\text{top}_{j'-2}(X_S)} \left( f_{X_i}(q) f_{X_j}(q') - f_{X_i}(q') f_{X_j}(q) \right) f_{X_S}(t_S) \, \mathrm{d}q' \, \mathrm{d}q \, \mathrm{d}t_S.$$

Now, define $\Delta = (1/3) \cdot \min_{k,l \in [m], k \neq l} |\theta_k - \theta_l|$. Then, the intervals $\{[\theta_k - \Delta, \theta_k + \Delta]\}_{k \in [m]}$ do not intersect. Further, there is a constant probability that $X_k$ is sampled from the interval $[\theta_k - \Delta, \theta_k + \Delta]$ is a positive constant for all $k \in S$ due to property (P3). Let us denote by $R$ the region in which this this happens for all $k \in S$. Hence, $\Pr[t_S \in R]$ is also a positive constant. We lower bound $\Omega$ by restricting the integration over $t_S$ to $R$. Further, due to property (P2), we have that over that region,

$$\beta = \min_{\substack{t_S \in R \\ q \in (\text{top}_{i'}(t_S), \text{top}_{i'-1}(t_S)] \\ q' \in (\text{top}_{j'-1}(t_S), \text{top}_{j'-2}(t_S)]}} f_{X_i}(q) f_{X_j}(q') - f_{X_i}(q') f_{X_j}(q)$$

is a positive constant. Hence, we get that $\Omega$ is lower bounded by a positive constant, as required.

**Case 2:** $j' = i' + 1$. This case is similar to the previous case, except that the conditions on $X_i$ and $X_j$ change slightly. In this case, we need to have $X_i, X_j \in (\text{top}_{i'}(X_S), \text{top}_{i'-1}(X_S)]$ along with $X_i > X_j$. Hence, the evaluation of $\Omega$ in Equation (11) changes to

$$\Omega = \int_{t_S} \int_{q = \text{top}_{i'}(X_S)}^{\text{top}_{i'-1}(X_S)} \int_{q' = \text{top}_{i'}(X_S)}^{q} \left( f_{X_i}(q) f_{X_j}(q') - f_{X_i}(q') f_{X_j}(q) \right) f_{X_S}(t_S) \, \mathrm{d}q' \, \mathrm{d}q \, \mathrm{d}t_S$$

$$\geq \int_{t_S} \int_{q = (2/3)\text{top}_{i'-1}(X_S)}^{\text{top}_{i'-1}(X_S)} \int_{q' = \text{top}_{i'}(X_S)}^{(2/3)\text{top}_{i'}(X_S) + (1/3)\text{top}_{i'-1}(X_S)} \left( f_{X_i}(q) f_{X_j}(q') - f_{X_i}(q') f_{X_j}(q) \right) f_{X_S}(t_S) \, \mathrm{d}q' \, \mathrm{d}q \, \mathrm{d}t_S$$

$$\geq \beta > 0,$$

where in the third transition,

$$\beta = \min_{\substack{t_S \in R \\ q \in \left[ (1/3)\text{top}_{i'}(t_S) + (2/3)\text{top}_{i'-1}(t_S), \text{top}_{i'-1}(t_S) \right] \\ q' \in \left[ \text{top}_{i'}(t_S), (2/3)\text{top}_{i'}(t_S) + (1/3)\text{top}_{i'-1}(t_S) \right]}} f_{X_i}(q) f_{X_j}(q') - f_{X_i}(q') f_{X_j}(q),$$

where $R$ is the same region as defined in Case 1. Once again, property (P2) implies that $\beta$ is a positive constant. Hence, $\Omega$ is lower bounded by a constant.

Hence, both the Thurstone-Mosteller and the Plackett-Luce models are PPM-$\alpha$ for a constant $\alpha > 0$. This completes the proof that the three classical noise models, under suitable assumptions, satisfy our four assumptions. $\qquad\qquad\qquad\qquad\qquad\qquad\qquad\qquad\qquad\qquad\quad$ $\square$ $\qquad\qquad\qquad$ $\square$

## F  Proof of Theorem 2

We first prove an upper bound on the accuracy of the uniform team. Fix an agent $i \in N$. Consider a state $s \in S$. For an alternative $a \in A_s$, $\Pr[f(\boldsymbol{\pi}_{is}^k) = a \mid \sigma_{is}]$ is the probability that the uniform team chooses $a$ when the biased truth is $\sigma_{is}$. The winner is chosen by applying $f$ to $k$ sampled votes from $P_i^2(\sigma_{is})$. Therefore, the neutrality of $P_i^2$ (assumption A1) and of $f$ (assumption in the theorem) imply that for every permutation $\tau$ of the alternatives,

$$\Pr[f(\boldsymbol{\pi}_{is}^k) = \tau a \mid \tau\sigma_{is}] = \Pr[f(\boldsymbol{\pi}_{is}^k) = a \mid \sigma_{is}]. \tag{12}$$

Consider the set of rankings $Good_{is} \subseteq \mathcal{L}(A_s)$ such that the true best alternative $\sigma_s^*(1)$ has the highest winning probability among all alternatives in $A_s$ if and only if $\sigma_{is} \in Good_{is}$. From Equation (12), we can see that permuting the alternatives in $\sigma_{is}$ permutes the winning probabilities of the alternatives accordingly. Hence, the rankings in $Good_{is}$ are obtained by taking one ranking in $Good_{is}$ and applying all possible permutations that fix (i.e., do not relabel) $\sigma_s^*(1)$, and thus do not change its position in the ranking. Thus, there exists a $k \in [m_s]$ such that $Good_{is}$ is the set of rankings where $\sigma_s^*(1)$ is in position $k$.

Let $Bad_{is} = \mathcal{L}(A_s) \setminus Good_{is}$. By assumption A2, there exists a constant $\eta > 0$ such that

$$\Pr[\sigma_{is} \in Bad_{is}] \geq \eta, \forall s \in S. \tag{13}$$

Further, when $\sigma_{is} \in Bad_{is}$, we know that there exists an alternative $a \in A_s$ with winning probability at least as high as that of $\sigma_s^*(1)$. Hence, the winning probability of $\sigma_s^*(1)$ is at most $1/2$. That is,

$$\mathbb{E}\left[X_{is}^k \mid \sigma_{is} \in Bad_{is}\right] \leq 1/2. \tag{14}$$

Putting everything together, we have that

$$\mathbb{E}\left[X_{is}^k\right] = \sum_{s \in S} \mu(s) \cdot \left[\Pr[\sigma_{is} \in Bad_{is}] \cdot \mathbb{E}\left[X_{is}^k \mid \sigma_{is} \in Bad_{is}\right]\right.$$
$$\left. + \Pr[\sigma_{is} \in Good_{is}] \cdot \mathbb{E}\left[X_i^k \mid \sigma_{is} \in Good_{is}\right]\right]$$
$$\leq \sum_{s \in S} \mu(s) \cdot \left[\Pr[\sigma_{is} \in Bad_{is}] \cdot \frac{1}{2} + \Pr[\sigma_{is} \in Good_{is}] \cdot 1\right] \leq \sum_{s \in S} \mu(s) \cdot \left[\eta \cdot \frac{1}{2} + (1 - \eta) \cdot 1\right]$$
$$= 1 - \frac{\eta}{2},$$

where the second transition holds due to Equation (14), and the third transition holds due to Equation (13). Taking $c = \eta/2$ proves the first part of the theorem.

For the results regarding the diverse team, recall that in a state $s \in S$, every agent $i \in N$ first draws its biased truth $\sigma_{is} \sim P_i^1(\boldsymbol{\theta}_s)$, and then draws a sample $\psi_{is} \sim P_i^2(\sigma_{is})$. Equivalently, we can say that each agent $i \in N$ draws its vote $\psi_{is} \sim (P_i^2 \circ P_i^1)(\boldsymbol{\theta}_s)$. Let $P_i = P_i^2 \circ P_i^1$. Thus, $\boldsymbol{\psi}_s^n$ is a profile consisting of one sample from $P_i(\boldsymbol{\theta}_s)$ for each $i \in N$. We want to show that $\lim_{n \to \infty} \Pr[f(\boldsymbol{\psi}_s^n) = \sigma_s^*(1)] = 1$. We establish this under each of the two conditions mentioned in the second part of the theorem.

First, let assumptions A1 and A3 hold, and let $f$ be a PD-c voting rule. Fix a state $s \in S$. Since $P_i^1$ and $P_i^2$ are PD-$\alpha$ (assumption A3), $P_i$ is PD-$\alpha^2$ due to Lemma 1. Thus, by definition we have that for every agent $i \in N$, every alternative $a \in A_s \setminus \{\sigma_s^*(1)\}$, and every $j \in [m_s - 1]$,

$$\Pr\left[\sigma_s^*(1) \in \psi_{is}([j])\right] \geq \Pr\left[a \in \psi_{is}([j])\right] + \alpha^2.$$

Recall that $T_{\boldsymbol{\psi}_s^n}(a, j)$ denotes the number of times alternative $a$ appears among first $j$ positions in $\boldsymbol{\psi}_s^n$. Due to Kolmogorov's strong law, we have that for all $a \in A_s \setminus \{\sigma_s^*(1)\}$ and for all $j \in [m_s - 1]$,

$$\lim_{n \to \infty} \Pr\left[T_{\boldsymbol{\psi}_s^n}(\sigma_s^*(1), j) \leq T_{\boldsymbol{\psi}_s^n}(a, j)\right] = 0.$$

Taking the union over $a \in A_s \setminus \{\sigma_s^*(1)\}$ and $j \in [m_s - 1]$, and applying the union bound, we get that the probability that $\sigma_s^*(1)$ is *not* the position-dominating winner in $\boldsymbol{\psi}_s^n$ is zero in the limit when $n \to \infty$. Since this holds for each state $s \in S$, it holds in expectation over the state space $S$. Moreover, the PD-c rule $f$ always outputs the position-dominating winner (with probability 1) whenever it exists. Hence, we have $\lim_{n \to \infty} \Pr[f(\boldsymbol{\psi}_s^n) = \sigma_s^*(1)] = 1$, as required.

The proof for Condorcet consistent rules is almost identical to the proof for PD-c rules. Let assumptions A1 and A4 hold, and let $f$ be a Condorcet consistent voting rule. Fix a state $s \in S$. Since $P_i^1$ and $P_i^2$ are PPM-$\alpha$ and PM-$\alpha$ respectively (assumption A4), $P_i$ is PM-$\alpha^2$ due to Lemma 2. Thus, by definition we have that for every agent $i \in N$ and alternative $a \in A_s \setminus \{\sigma_s^*(1)\}$,

$$\Pr[\sigma_s^*(1) \succ_{\psi_{is}} a] \geq \Pr[a \succ_{\psi_{is}} \sigma_s^*(1)] + \alpha^2.$$

Due to Kolmogorov's strong law, a majority of votes in $\boldsymbol{\psi}_s^n$ would rank $\sigma^*(1)$ above $a$ with probability 1 as $n \to \infty$, for every $a \in A_s \setminus \{\sigma_s^*(1)\}$. Hence, in the limit, $\sigma_s^*(1)$ becomes the Condorcet winner with probability 1. Since this holds for each state $s \in S$, it holds in expectation over the state space $S$. Moreover, the Condorcet consistent voting rule $f$ must output the Condorcet winner (with probability 1) whenever it exists. Hence, once again we have $\lim_{n \to \infty} \Pr[f(\boldsymbol{\psi}_s^n) = \sigma_s^*(1)] = 1$, as required. $\qquad\qquad\square$(Proof of Theorem 2) $\qquad\square$

While Theorem 2 is already quite general, it is possible to generalize it even further. We preferred to present the simpler version for ease of exposition; but let us informally say that it is also possible to handle the case where the ground truth is an objective true quality for each alternative (and then one would rather know the expected quality of the chosen alternative), and the case where the number of alternatives in each state is not fixed.

## G  Parametrized Agents

In Table 1 we present the parameters that were sampled to generate parametrized versions of Fuego. For each random draw, we used a uniform random distribution, defined in the interval shown in the column "Range". Also, depending on the domain of each parameter, we sample integers or floating point numbers. A detailed description of these parameters is available in the Fuego documentation, at `http://fuego.sourceforge.net/fuego-doc-1.1/`.

## H  Additional Experimental Results

In this section we present further analysis of our experimental results. First we show that the original Fuego is, indeed, stronger than the parametrized agents. Like in our experiments, we ran 1000 $9 \times 9$ Go games, with the system under evaluation playing as white, against the original Fuego playing as black. In Figure 3 we can see the winning rate of Fuego and of each one of the parametrized agents. The original Fuego is the strongest agent (with $p < 0.01$ for all but 3 agents), having a winning rate close to 50%. The parametrized agents, on average, have a winning rate of 32.3% (std: 10.4%).

We also evaluate the diversity of a team of parametrized agents, by analyzing a sample of 10 parametrized agents. We use the metric proposed in [19], where diversity is defined as the average Hellinger Distance [11] between the probability density functions (PDFs) of all possible combinations of pairs of agents across different world states. We show three different results: *Control*

Figure 3: Winning rate of Fuego and of the parametrized agents.

| Parameter | Domain | Range |
|---|---|---|
| uct_param_globalsearch mercy_rule | Integer | [0,1] |
| uct_param_globalsearch territory_statistics | Integer | [0,1] |
| uct_param_globalsearch length_modification | Float | [0, 0.5] |
| uct_param_globalsearch score_modification | Float | [0,0.5] |
| uct_param_player forced_opening_moves | Integer | [0,1] |
| uct_param_player reuse_subtree | Integer | [0,1] |
| uct_param_player use_root_filter | Integer | [0,1] |
| uct_param_policy nakade_heuristic | Integer | [0,1] |
| uct_param_policy fillboard_tries | Integer | [0, 5] |
| uct_param_rootfilter check_ladders | Integer | [0,1] |
| uct_param_search check_float_precision | Integer | [0,1] |
| uct_param_search prune_full_tree | Integer | [0,1] |
| uct_param_search rave | Integer | [0,1] |
| uct_param_search virtual_loss | Integer | [0,1] |
| uct_param_search weight_rave_updates | Integer | [0,1] |
| uct_param_search bias_term_constant | Float | [0, 1.0] |
| uct_param_search expand_threshold | Integer | [1,4] |
| uct_param_search first_play_urgency | Integer | [1,10000] |
| uct_param_search knowledge_threshold | Integer | [0,10000] |
| uct_param_search number_playouts | Integer | [1,3] |
| uct_param_search prune_min_count | Integer | [1,128] |
| uct_param_search randomize_rave_frequency | Integer | [0,200] |
| uct_param_search rave_weight_final | Integer | [1000,10000] |
| uct_param_search rave_weight_initial | Integer | [0,999] |

Table 1: Parameters sampled to generate different versions of Fuego.

compares each agent with a second sample of itself, in order to measure the noise in our evaluation; *Parametrized Agents* compares all possible pairs of parametrized agents, in order to estimate the diversity of our team; and *Independent Agents* compares each parametrized agent with Pachi [4], an independently developed Computer Go program. In order to perform the analysis, we estimate the PDFs of Pachi and 10 agents from the diverse team, using 100 different board states. For each board state we sample 100 moves for each agent. The results are shown in Figure 4(a). These results indicate that the level of diversity is especially high when the parametrized agents are compared with Pachi, suggesting that the current parametrization methodology falls short of creating an idealized diverse team. That said, the methodology does lead to some diversity, as indicated by the statistically significant difference between the Control bar and Parametrized Agents bar.

We also evaluate the level of diversity by testing whether there is a set of board states where all parametrized agents have a low probability of playing the best action. Again, we evaluate a sample of 10 agents from the diverse team. We first estimate the best move for each of 100 board states. To this end, we use Fuego to evaluate the given board state, but with a time limit 50x higher than the default one. Then, based on the previous estimated PDFs of the parametrized agents, we can obtain the probability of each agent playing the optimal action. Finally, we calculate the proportion of board states in which all parametrized agents play the best action with probability below a certain threshold. The results are shown in Figure 4(b). It turns out that all parametrized agents play the optimal action with probability smaller than $1/2$ in 40% of the board states. Moreover, in 10% of the board states, the probability of playing the optimal action is lower than 10%. Hence, there is still a large set of board states in which all agents play badly, regardless of the parametrization.

(a) Diversity of the parametrized agents, compared with a second sample and with the diversity between independently developed agents

(b) Percentage of world states where all parametrized agents have probability of playing the best action below the given threshold

Figure 4: Evaluation of the diversity of the parametrized agents, and the fraction of states in which all of them have a low probability of playing the optimal action. The error bars show 99% confidence intervals.

## Footnotes

[1] Note that due to neutrality of $P^1$ and $P^2$, these probabilities are independent of the true ranking.