[Reviews · NeurIPS 2014]

Submitted by Assigned_Reviewer_7

The paper considers multi-agent systems where a voting rule is used to aggregate decisions of individual agents to solve a problem. One can consider the agents as sampling from a noisy distribution centered around the true ranking of the alternatives at each state. They show that if the agents are copies of each other, in the sense that the agents produce samples (preferences for alternatives) from the same distribution then it is likely that system votes for a sub-optimal alternative. On the other hand, if we have many agents that have sufficient diversity then the system is likely to vote for the optimal alternatives in almost every state. The paper then concludes with an experimental study of agents playing Go games.

The paper is clear and well written, however, it was not clear to the reviewer what the new contributions are over existing results which qualitatively resemble this paper's results (such as Marcolino's work that they refer to). It seems that the main difference in the setting is that agents can give a preference ordering for alternatives at each stage rather than vote for a single alternative.

Minor: In the experimental section of the paper diverse agents are generated by varying parameters in an existing Go playing agent. From the perspective of a reader unfamiliar with the details of the system, it is not obvious that such parametrization will indeed result in a diverse set of agents. The paper does discuss this issue in Appendix H and towards the end of section 4, it would be better if this justification (or reference to Appendix H) is added earlier when the agents are introduced.
Summary: The results of the paper are interesting but a more detailed comparison to related work is needed.

[Revised, following author feedback].

Submitted by Assigned_Reviewer_21

The paper provides a framework for analyzing the quality of a vote among a team of agents (compared to some underlying true ranking) where the teams are generated differently: by multiple instantiations of the same agent ("Uniform"), and by a single instantiation of several different types of agents ("Diverse"). Under some conditions, the authors show that as the number of agents grow the Diverse team converges in probability almost surely to choosing the correct choice, where the Uniform team always has a non-zero probability of making a mistake.

The paper is quite clear and the technical quality is high. The results are significant and relevant. The main idea here is that voting among a diverse team leads to robust choices, whereas a population of fixed players does not have enough variance and will play "too predictably", but the authors provide a nice theoretical foundation to quantify these notions and show that even diversity obtained from a single (widely-used) Go program with different parameters can lead to significant performance boosts in 9x9 Go (against the same copies of the optimally-tuned parameter version of the Go program) when using an appropriate voting scheme. The authors also show how their assumptions and voting rules relate to other rules used in the literature.

I have a some problems with the paper. The first is that there seems to be a disconnect between the experiments and the foundations that came before it. The experimental section is specific, focusing on the single application to Go, while the theoretical framework is quite general. I was expecting to see experiments comparing different noise models, something that validates the assumptions listed in section 3, or some stronger link between the first half of the paper and second. As far as I can tell there is no mention of whether the Go application even satisfies the assumptions 1-4, so does Theorem 2 even hold in this experiment? This is also true for the discussion section, which seems to again focus on the application to games while the first part is more general.Also, as this is follow-up work based heavily on past work ([18,19]) the authors should do a better job of highlighting what is new in this paper. The noise model approach certainly seems appropriate for the computer Go setting where there indeed is some underlying true ranking of alternatives, but this might not be true in other more general social choice settings such as elections.. can anything be said on whether this approach could be applied when the existence of a true ranking is unknown?
Summary: This paper provides a significant theoretical result that established the asymptotic optimality of a team of diverse agents (under certain appropriate assumptions) while showing suboptimality of a team of uniform agents. The experimental results, while significant, feel somewhat disconnected from the main results and the originality compared to previous work could be more clear.

Submitted by Assigned_Reviewer_34

In this paper, the author propose a novel two-stage noisy voting model and theoretically demonstrated that a uniform team must make some mistakes, whereas a diverse team converges to perfection as the number of agents grows. The result of the experiment, which was conducted in the Computer Go domain, is consistent with the theoretical analysis.

This paper is well-written, and the results are solid and interesting. This work has both theoretical and methodological contributions:
1.Prove diverse vote can beat uniform vote under their proposed restrictions on noise models, i.e., PD-alpha, PM-alpha, PPM-alpha. Furthermore, they show these assumptions in the theorem is mild by demonstrating that the three classical noise model satisfy all assumptions. As a result, the theorem can be widely used.
2.In the experiment, authors automatically generating arbitrarily many diverse agents and extracting rankings of moves from algorithms, are practical.

Following are concerns about this paper:
1. The novelty of this paper: Please make a clear statement on the technical contribution of this paper comparing to Marcolino et. al. 2013.

2. The experiment: although authors explains that there may be gap between figure (a) and the main theorem, the obvious winning rate drop for diverse+Copeland is somehow indicating the performance of diverse voting is not robust under some voting rule. A more comprehensive experiment results will offer more insights to this problem. Authors said that it is time-consuming to present diverse + all voting rules. Please give more explanations to the computation cost, and it would be better if the authors try to conquer it by some tricks.

3. For Go, the state are dependent but not i.i.d.. The setting of the main results of this paper is not suitable for Go.

Minor comments: 1. Please indicate the domain of phi in Mallows model. 2. Add one bullet for Plackett-Luce model. 3. “k” is used to indicate positions. Please use another notation for copy number in uniform vote.
Summary: This paper is well-written, and results is interesting. Authors need to state their technical contribution with a very related reference.
Author Feedback
Author rebuttal: RESPONSE TO ALL REVIEWERS:

A central concern raised by all three reviewers is the comparison with [18,19]. We decided to omit a detailed comparison due to space constraints, and while, in retrospect, this was a poor decision, it is one that is easy to correct. Below we compare our work to [18,19]; we also plan to flesh out the comparison in the paper.

Executive summary:

Our paper is fundamentally different from the papers by Marcolino et al. [18,19]. Our "double error" model for generating noisy rankings, which builds on ideas from computational social choice and machine learning, is conceptually novel, and significantly more nuanced than the theoretical models of Marcolino et al. In particular, it allows us to reason about a wide variety of ranked voting rules, whereas Marcolino et al. only study one simple rule (plurality). Moreover, our experimental analyses focus on the comparison of ranked voting rules (again, in contrast to Marcolino et al.), and demonstrate how the performance of different teams change as the number of agents grows larger. In order to do so, we develop and apply novel techniques for generating diverse agents and extracting accurate rankings from these agents. In summary, our theory and experiments are not just more general or quantitatively better than those of Marcolino et al. — they are qualitatively different.

Details:

Model and results of [18]: On a theoretical level, [18] models each agent as a probability distribution over actions, which depends on the state. Plurality aggregates the "votes" of multiple agents by choosing the most popular action. Their main result states that a diverse team can outperform the uniform team *only if* one of the agents in the former team is more likely to choose the correct action than the agent of the latter team, in *some* state. [18] Does not present any theoretical results that quantify the benefit of diversity. Concerning the experimental results, [18] only presents results with 4 and 6 agents, and shows the performance of the two teams under plurality voting.

Model and results of [19]: [19] focuses on explaining an observed improvement of the diverse team in Go, as the board size grows larger. On a theoretical level, [19] studies a model that is very similar to [18], that is, agents are modeled as probability distributions over actions, and plurality voting is used to aggregate votes; but [19] allows the number of actions to grow. Concerning the experimental results, [19] again only presents results with 4 and 6 agents and plurality voting, and focuses on the impact of board size.

Comparison with our paper:

- Our theoretical model is completely different, significantly more nuanced, and requires a much more careful analysis. First, votes are rankings rather than single actions. Second, a simple generalization of [18] and [19] would assume a distribution over rankings, but, instead, our "double error" model captures a wide variety of ways in which noisy rankings can be generated, drawing on a significant body of literature in social choice and ML. Crucially, our main theoretical result (Theorem 2) also captures a wide range of voting rules, in contrast to [18] and [19], which only study plurality voting.

- Our empirical results focus on comparing different ranked voting rules, and analyzing the performance of the two types of teams as the number of agents grows larger: we show results for 5 different voting rules and up to 15 agents. These new results are only possible because of two novel techniques. First, we devise a procedure for systematically generating diverse agents, which is crucial for understanding whether the asymptotic convergence results hold for teams of realistic size. In contrast, Marcolino et al. are limited to off-the-shelf, open-source Go agents, and hence only consider up to six agents. Second, we design a procedure for extracting rankings from agents that are originally designed to output a single action, which is required to generate the inputs to ranked voting rules such as Copeland and Borda count. While we view this as a less important point, it is worth noting that our results are also quantitatively better: we get close to 90% winning rate against Fuego, by using the Copeland voting rule and a team of 10 diverse agents, whereas if we use plurality (the voting method used by [18]), we get less than 50% winning rate under the same setting (see Fig 1(a) in the paper).

RESPONSE TO REVIEWER 1:

The reviewer asks whether noise models are appropriate in more general social choice settings such as elections, where there may not be a true ranking. The results cannot be applied when there is no true ranking. Crucially, though, current research in computational social choice focuses on elections where there *is* a true ranking. These elections arise in crowdsourcing and human computation contexts. Just to give one example, in the citizen science game EteRNA, a voting procedure is used to select RNA molecule designs (submitted by players) to synthesize in a lab; the true quality of a design is a stability score that it receives after it is synthesized, that is, there is a true ranking of designs by their stability. For more details, see [8,23].

The reviewer also asks whether the assumptions of Theorem 2 hold in our experiments in the Go domain. Theorem 2 has two types of assumptions, on the noise models and on the voting rule. The assumptions on the voting rule definitely hold: All voting rules in Figure 1(b) are neutral (required in part 1), and they are all either PD-c or Condorcet consistent (required in part 2). As for the assumptions on the noise model, it is hard to say what the most appropriate noise models for the Go domain are, but we try to make the theorem as general as possible by imposing rather mild assumptions. The assumptions required in the first part of the theorem, in particular, are extremely mild and capture any "reasonable" noise model.